# Growth Plate Chondrocytes: Skeletal Development, Growth and Beyond

**DOI:** 10.3390/ijms20236009

**Published:** 2019-11-29

**Authors:** Shawn A. Hallett, Wanida Ono, Noriaki Ono

**Affiliations:** Department of Orthodontics and Pediatric Dentistry, University of Michigan School of Dentistry, Ann Arbor, MI 48109-1078, USA; shallett@umich.edu (S.A.H.); wono@umich.edu (W.O.)

**Keywords:** chondrocyte, osteoblast, endochondral ossification, growth plate, parathyroid hormone-related protein, hedgehog, skeleton, regeneration

## Abstract

Growth plate chondrocytes play central roles in the proper development and growth of endochondral bones. Particularly, a population of chondrocytes in the resting zone expressing parathyroid hormone-related protein (PTHrP) is now recognized as skeletal stem cells, defined by their ability to undergo self-renewal and clonally give rise to columnar chondrocytes in the postnatal growth plate. These chondrocytes also possess the ability to differentiate into a multitude of cell types including osteoblasts and bone marrow stromal cells during skeletal development. Using single-cell transcriptomic approaches and in vivo lineage tracing technology, it is now possible to further elucidate their molecular properties and cellular fate changes. By discovering the fundamental molecular characteristics of these cells, it may be possible to harness their functional characteristics for skeletal growth and regeneration. Here, we discuss our current understanding of the molecular signatures defining growth plate chondrocytes.

## 1. Introduction

Growth and development of the axial and appendicular skeletons is a complex and multifactorial process tightly regulated by numerous signaling pathways. Most bones in the vertebrate skeleton are formed through endochondral ossification, in which an initial cartilage template is systematically replaced by bones [1]. Chondrocytes, which are derived from undifferentiated mesenchymal cells in condensations, serve to both drive the growth of the skeletal elements and to form a scaffold for the subsequent mineralization by osteoblasts [1]. Not only do chondrocytes provide the fundamental basis for proper skeletal development, their functions extend to support bone remodeling and regeneration during fracture healing. Recent applications of in vivo lineage-tracing approaches, single cell transcriptomics and various other omic technologies are beginning to open the avenue for elucidating the molecular properties of growth plate chondrocytes [2,3,4]. In this review, we will describe the fundamental properties and molecular regulators of growth plate chondrocytes and their roles in endochondral bone formation.

## 2. Chondrocyte Function in Endochondral Ossification

### 2.1. Principles of Bone Formation and Elongation

Endochondral ossification begins with the condensation of undifferentiated mesenchymal cells [5]. The cellular components of this condensation differentiate into chondrocytes that lay down the cartilage matrix, which is eventually replaced by bone. Hypertrophic chondrocytes instruct adjacent perichondrial cells to differentiate into osteoblasts, which secrete an extracellular matrix and form the periosteal bone collar [6]. Shortly thereafter, the primary ossification center (POC) forms inside the cartilage template. This is done by hypertrophic chondrocytes attracting blood vessels into the center of the cartilage template, which leads to the formation of a highly vascularized endosteum within the nascent marrow space stimulated primarily by vascular endothelial growth factor (VEGF) [7]. Importantly, hypertrophic chondrocytes also express the molecular signals that direct the mineralization of the surrounding matrix, such as Indian hedgehog (Ihh). Subsequently, hypertrophic chondrocytes undergo apoptosis or (trans)differentiate into osteoblasts to form a bony matrix known as the primary spongiosa, which is the future site of trabecular bone [8]. Importantly, bone elongation is dictated by the rate at which proliferating chondrocytes undergo hypertrophy [9]. At the epiphysis, the secondary ossification center (SOC) forms through successive rounds of chondrocyte hypertrophy, vascular invasion and osteoblast differentiation until the cartilage templates are replaced by bone, dividing the pre-existing cartilage into the growth plate [10,11]. While the latter articular cartilage is permanent and continues to function at the joint surface, the former growth cartilage is transient and eventually disappears in humans (but not in mice) when the skeleton reaches a fully mature state. The articular cartilage develops from a different cell origin and through a different mechanism from those of growth plate chondrocytes [11,12]. Hypertrophic chondrocytes also instruct multinucleated bone-resorbing osteoclasts to remove the cartilage template, which in turn enables osteoblasts to use the remnants of this matrix as a scaffold for the deposition of bone matrix. Bone elongation continues throughout puberty and well into early adulthood. Eventually, the growth plate cartilage is completely replaced by bones in humans, thus underscoring its transient nature.

In the postnatal growth plate, there are three layers of chondrocytes that contribute to the longitudinal growth of long bones. The resting zone is located at the top of the growth plate and maintains the integrity of the growth plate structure, particularly by expressing parathyroid hormone related protein (PTHrP). This zone is characterized by being the farthest from the ossification front of the POC and by the indistinctive morphology of the chondrocytes embedded within the cartilaginous matrix [13]. As cells within the resting zone begin to divide, the proliferating zone is formed. This zone is characterized by the organization of chondrocyte clones that arrange themselves into columns, contributing to bone elongation via interaction with adjacent cells. An intermediary zone termed the pre-hypertrophic zone lies between the proliferating zone and the hypertrophic zone, where Ihh is most abundantly expressed.

Hypertrophic chondrocytes are generated from the terminal differentiation of chondrocytes in the proliferating zone and are characterized by an enlarged cell type organized into columns and parallel to the axis of elongation [14]. The enlargement of hypertrophic chondrocytes is the major contributing factor regulating the growth rate in endochondral bones. Furthermore, this metric is largely responsible for the variations in skeletal growth rates amongst endochondral bones within an individual and between homologous skeletal elements in similar species. There are three phases of hypertrophic cell enlargement, classified by intracellular volume increases in response to swelling [9]. Cell swelling during chondrocyte hypertrophy enables chondrocytes within the growth plate to enlarge rapidly while lowering their energetic cost of growth.

The hypertrophic zone interacts with the resting zone through a PTHrP-Ihh negative feedback loop that promotes the formation of columnar chondrocytes and simultaneously maintains growth plate structure [15]. The fine-tuned molecular interactions of growth plate chondrocytes are instrumental in orchestrating postnatal bone elongation and, possibly, skeletal regeneration. To date, our understanding of the fundamental biological mechanisms that regulate these processes is not completely developed. Here, we will further discuss the current state of knowledge regarding the normal development and growth of the skeletal system.

### 2.2. Investigation of Putative Role of Sox9 in Early Mesenchymal Condensation Formation and Growth Plate Chondrocyte Differentiation

SRY-Box 9 (Sox9) is a transcription factor expressed by cells in mesenchymal condensations and proliferating chondrocytes [16]. Sox9 is expressed by osteo-chondroprogenitors, as *Sox9-cre* allele-based fate mapping labels all chondrocytes and osteoblasts [17]. *Sox9*-deficient undifferentiated cells do participate in condensation formation but lack the ability to form chondrocytes [17]. Functionally, *Sox9* deletion in early limb mesenchymal cells lack the formation of cartilaginous condensations associated with increased cellular apoptosis [18]. Therefore, Sox9 is a master regulator of mesenchymal cell condensation and chondrocyte differentiation; it also functions during the transition from proliferative to hypertrophic chondrocytes. Studies have shown that Sox9 is an upstream activator of a number of cartilage matrix genes, such as type 2 collagen alpha 1 chain (Col2a1), type 11 collagen alpha 2 chain (Col11a2) and aggrecan (Acan). Cells expressing these genes coincide with precursor cells for multiple mesenchymal lineages including chondrocytes, osteoblasts, stromal cells and adipocytes, which provide these descendants for over a year in mice [19]. Importantly, given the diverse differentiation potential and persistence of *Col2a1*, *Sox9* and *Acan*-expressing cells in bones, these cells may also provide the molecular cues that activate osteoblast differentiation during explosive osteogenesis associated with continuous longitudinal bone growth.

## 3. Understanding the Differential Roles of the Three Zones of the Postnatal Growth Plate

### 3.1. Evaluating the Stem Cell-Like Properties of Resting Zone Chondrocytes

The resting zone is located at the top of the postnatal growth plate, immediately adjacent to the SOC. Early studies of resting zone chondrocytes describe these cells possessing stem-like qualities due to their infrequent cellular division and the prediction that they can feed their daughter cells into the adjacent proliferating zone [20,21]. Chondrocytes of the resting zone have been suggested to clonally contribute to the formation of the proliferating zone as functional and mechanical units, orchestrating longitudinal growth of bones in the appendicular and axial skeletons [13]. The proliferating and hypertrophic zones were surgically removed in rabbit growth plates in order to better elucidate the stem cell-like qualities of resting chondrocytes. In some of the samples, a complete growth plate regenerated, as indicated by the presence of all layers of chondrocytes, vascular invasion at the metaphyseal edge of the hypertrophic zone and signs of active osteogenesis in the primary spongiosa. In other samples, only proliferative zones regenerated, but the hypertrophic zone was absent with the growth plate being disorganized [13]. This study shed light on resting chondrocytes as a fundamentally important cell type for the development and organization of the growth plate. In the same study in rabbits, in order to separate the resting zone from the proliferating zone, the resting zone was surgically excised and reimplanted in an ectopic site lying perpendicular to the long axis of the bone. When reimplanted, the resting zone promoted the proper formation of columns of proliferative chondrocytes and gradually reestablished the growth plate in a time-dependent manner. They further conducted auto-transplantation studies by surgically removing and reorienting the resting zone alongside columns of proliferative and hypertrophic chondrocytes. In 19 of 21 growth plates, proliferative chondrocytes had undergone a 90-degree shift to realign themselves perpendicular to the pre-existing resting zone and had also established a normal morphology. Proliferative zone chondrocytes that lie farther from the auto-transplanted resting zone did not undergo such a shift, demonstrating the importance of spatial relationships with the resting zone. In contrast, growth plate chondrocytes adjacent to the ectopic resting zone did not undergo hypertrophic differentiation, thereby inhibiting further ossification of the adjacent proliferative zone cartilage [13].

Therefore, chondrocytes in the resting zone have two distinct functions dictating the postnatal growth plate: stem cell-like properties as well as the ability to coordinate the proper differentiation into proliferative and hypertrophic chondrocytes in a non-cell autonomous manner, at least in rabbits. Resting zone chondrocytes produce a growth plate-orienting factor that instructs the proper spatial orientation of adjacent proliferative chondrocytes. These foundational experiments provide the first evidence for the importance of resting zone chondrocytes in the proper formation, maintenance and, possibly, regeneration of the growth plate, establishing the resting zone as a functional cellular niche of the developing postnatal growth plate.

### 3.2. Molecular Properties of Resting Zone Chondrocytes

The resting zone of the growth plate has been now identified to act as a reservoir for quiescent skeletal stem cells, which is important for promoting the elongation of bones undergoing endochondral ossification [2,3]. In the first study, a *Pthrp-mCherry* knock-in reporter allele mouse model was generated to dynamically assess the characteristics of PTHrP^+^ cells in normal tissues. At 6 days post-natal development (P6), PTHrP^+^ chondrocytes appear in the resting zone of the growth plate. Just 3 days later, PTHrP^+^ cells increase significantly within the central portion of the resting zone, marking small cells mostly devoid of cellular proliferation. Interestingly, this increase in PTHrP^+^ chondrocytes in the resting zone coincided with the formation of the SOC, suggesting a fundamental relationship between SOC formation and activation of PTHrP. Furthermore, these PTHrP^+^ cells express a panel of immunophenotypically defined cell-surface markers for transplantable skeletal stem cells and progenitors, marked as CD45^−^Ter119^−^CD31^−^CD51^+^CD90^−^CD105^−^CD200^+^, suggesting that PTHrP^+^ cells might possess skeletal stem cell-like properties with robust ex vivo clonal activities [2,22].

Furthermore, a tamoxifen-inducible *Pthrp-creER* line was generated to achieve an in vivo lineage analysis of PTHrP^+^ chondrocytes in the resting zone. After tamoxifen injection at P6 and following 6-days of chase, *Pthrp-creER* marks a subset of centrally located PTHrP^+^ resting chondrocytes within the resting zone. After 12 days of chase at P18, these cells generate short columnar chondrocytes (<10 cells) within the proliferative zone. Following one month of chase at P36, these cells form much longer columnar chondrocytes originating from the resting zone (>10 cells) that reach the hypertrophic zone. These cells thereafter become osteoblasts in the primary spongiosa, showing that PTHrP^+^ resting chondrocytes have the ability to clonally differentiate into multiple cells types within the growth plate and beyond, including proliferative and hypertrophic chondrocytes as well as osteoblasts in the primary spongiosa [2].

In the latter study, the authors discover a stem cell niche within the postnatal growth plate using *Col2a1-creER-Brainbow2.1* multicolor lineage tracing and computational modeling [3]. Using this model, they demonstrate the existence of chondrocyte precursor cells at as early as embryonic day 14.5, that undergo a shift in clonality, initially forming multiclonal short columns in the proliferative zone through a consumption program and later forming monoclonal long columns through a self-renewal program. Interestingly, this clonality shift coincides with the development of the SOC. This cellular population also exhibits the ability to replicate asymmetrically and differentiate in a unidirectional manner from a small number of chondrocytes in the resting zone to clonal cells arranged into columns. Further functional analyses of the Tsc1 signaling pathway show that inhibition of *Tsc1*, which is a negative regulator of mTORC1 activity, in chondrocytes causes obvious abnormalities in the resting zone, indicated by the inability of resting chondrocytes to differentiate into columnar chondrocytes within the proliferating zone [3,23].

With regard to regeneration, PTHrP^+^ resting chondrocytes migrate into the drill-hole injury site within the growth plate and become osteoblasts, demonstrating that these cells acquire osteoblast-like properties in response to injury [2]. To assess the functional importance of PTHrP^+^ resting chondrocytes, inducible partial cell ablation experiments using an inducible diphtheria toxin fragment A allele (iDTA) was conducted. Following tamoxifen injection, the height of each layer of the growth plate was altered in response to DTA-mediated ablation of PTHrP^+^ chondrocytes in the resting zone, associated with reduction of the proliferating zone and significant expansion of the hypertrophic and resting zones. Therefore, partial loss of PTHrP^+^ resting chondrocytes is sufficient to alter the architecture of the growth plate by inducing premature chondrocyte hypertrophy in the proliferating zone [2]. These lineage-tracing, functional conditional knockout assays and skeletal regeneration studies demonstrate that PTHrP^+^ resting chondrocytes are a dedicated source of columnar chondrocytes in the growth plate by providing a forward PTHrP-mediated signal to transit-amplifying progeny in a non-cell autonomous manner, and that they are a fundamental driver for the explosive skeletal growth and bone elongation that occurs in mammals.

### 3.3. Assessing the Switch from Chondrocyte Hypertrophy to an Osteoblast-Like State

The molecular regulation of the differentiation of hypertrophic chondrocytes into osteoblasts is not well understood. Although chondrocyte and osteoblast lineages are derived from a common ancestor cell type, both regulated primarily by the transcription factors SRY-Box 9 (Sox9), runt-related transcription factor 2 (Runx2) and Sp7 (i.e., osterix, Osx), these lineages are typically regarded as separate once they diverge from a bipotential osteo-chondroprogenitor state. However, the regulators of these two cell types are known to have overlapping roles. For example, Sp7 (Osx) is genetically downstream of Runx2, both expressed in both prephypertrophic chondrocytes and preosteoblasts, and is required for proper differentiation into hypertrophic chondrocytes and osteoblasts [24]. Although *Osx*-expressing cells invade into the cartilage template along with invading blood vessels during endochondral ossification to become both osteoblasts and stromal cells in the marrow space [18,19], it is possible that hypertrophic chondrocytes can also differentiate into these cells in an Osx-dependent manner. Given that the maintenance of bone is executed via the renewal of osteoblasts, this aspect is of particular significance for clinical therapeutics.

One well-accepted marker of hypertrophic chondrocyte is *Col10a1*. A *Col10a1-cre* mouse model was crossed with a *Rosa26R* reporter that encodes a β-galactosidase (*RLacZ*) reporter construct (*Col10a1-cre; RLacZ*) [8]. Interestingly, *Col10a1* expression is limited to hypertrophic chondrocytes in the developing endochondral bones at E15.0, without being expressed within the bone collar. However, just half a day later, these cells begin to invade into the POC. Interestingly, at 3 months of age, cells descended from initially Col10a1^+^ cells are present at the chondro-osseous junction beneath the growth plate, suggesting that Col10a1^+^ hypertrophic chondrocytes can commit to an osteogenic fate in adulthood and may thrive as osteocytes. Transgenic overexpression of mutant Col10a1 results in endoplasmic reticulum stress and the failure of hypertrophic chondrocytes to terminally differentiate, resulting in delayed endochondral bone formation and a chondrodysplasia phenotype [25]. These studies redefine the concept that chondrocytes and osteoblasts are separate lineages by establishing a link between the two cell types during skeletal development.

To assess the lineage specification changes of *Col10a1*-expressing hypertrophic chondrocytes, a tamoxifen-inducible *Col10a1-creER* line was generated. Similar to the *Col10a1-cre; RLacZ* model, the *Col10a1-creER* line marked hypertrophic chondrocytes. Interestingly, after 36 h of chase, *Col10a1-creER*-marked cells overlapped with Osx^+^ and Col1a1^+^ cells, suggesting that Col10a1^+^ hypertrophic chondrocytes might give rise to preosteoblasts and osteoblasts [8]. During further chase, these cells persist throughout the cortical and trabecular bone into the marrow cavity. Moreover, these cells became a small number of osteoblasts and osteocytes embedded within the matrix, marked by sclerostin [26]. Therefore, once committed, lineage-specified hypertrophic chondrocytes can differentiate into preosteoblasts, fully mature osteoblasts and osteocytes embedded within the matrix. Importantly, this cell type persists into adulthood and expresses markers of fully differentiated osteoblasts, indicating its potential role in postnatal long bone development and growth.

To determine the functional role of Col10a1^+^ cells in bone repair and remodeling, hypertrophic chondrocytes were isolated from a *Col10a1-cre;RLacZ/yellow fluorescent protein (YFP)* reporter and were grafted into the repair sites of drill hole surgery. Interestingly, by 8 days post-operation, these transplanted cells became osteoblast and osteocyte-like cells in the repair site. Another report has identified similar results using a *Col10a1-cre* mouse model to label hypertrophic chondrocytes and suggests the potential for this cell type to undergo ‘transdifferentiation’ prior to committing to an osteoblast-like state [27]. Constitutively active *Col10a1-cre* and tamoxifen inducible *Acan-creER* mouse models show that descendants of Col10a1^+^ cells are present in the trabecular bone and within the bone matrix at later stages of postnatal bone development and express typical osteoblast markers. Furthermore, chondrocytes marked by *Acan-creER* undergo ‘transdifferentiation’ into non-chondrocytic cells present within the primary spongiosa. However, there is insufficient evidence suggesting that *aggrecan* expression is restricted to hypertrophic chondrocytes and may be more representative of additional cell types [28]. Taken together, postnatal hypertrophic chondrocytes, which may be predominantly labeled by *Col10a1-cre*, can commit to an osteogenic fate during normal endochondral bone formation, regeneration and remodeling [8,27].

These studies establish a prevailing view that hypertrophic chondrocytes can take on an osteogenic fate during normal endochondral ossification and bone regeneration. Moreover, they suggest that Col10a1^+^ cells have the potential to undergo multiple cell fates, other than just becoming osteoblasts. More specifically, these studies show that hypertrophic chondrocytes can become osteoblasts and osteocytes, contribute to trabecular and cortical bone in addition to undergoing apoptosis. Results discussed here introduce a revision in the concept regarding the origin of osteoblasts in endochondral bones.

### 3.4. Borderline Chondrocytes in the Developing Growth Plate

Early studies classified borderline chondrocytes as being located directly underneath the perichondrium of pre-invasion cartilage rudiments and eventually at the border of the post-natal growth plate [29]. Recent lineage-tracing studies utilizing *Pthrp-creER* transgenic mouse models have confirmed the presence of borderline chondrocytes at the periphery of the neonatal growth plate [4]. Single cell RNA-sequencing analysis was used to elucidate the molecular characteristics of neonatal chondrocytes harvested from the growth plate. Interestingly, a cluster of borderline chondrocytes is identified, which is not enriched for the hypertrophic chondrocyte markers, *Col10a1* and *Sp7*. To determine the functional contribution of borderline chondrocytes, lineage-tracing experiments were conducted using *Pthrp-creER*. Following tamoxifen injection at P0, a small population of borderline chondrocytes is labeled, with almost all of them expressing *Col2a1* and *aggrecan*, while only a small percentage of them expressing *Col10a1*. These cells have the ability to translocate into the marrow space passively without actively proliferating, as they are resistant to EdU incorporation, and without undergoing active cell apoptosis. After two weeks of chase, these neonatal borderline chondrocytes move into the marrow space and become stromal cell-derived factor 1 (*Cxcl12*)-expressing bone marrow stromal cells and osteoblasts. These cells further persist in these locations after 36 days of chase, particularly on the trabecular bone surface, but eventually disappear after 67 days of chase. Therefore, neonatal borderline chondrocytes behave as a transient mesenchymal precursor for osteoblasts and marrow stromal cells during early postnatal bone growth [4].

## 4. Dissecting the Molecular Properties of Growth Plate Chondrocytes

### 4.1. Ihh Regulation in the Developing Growth Plate

The Hedgehog family of proteins comprises three separate members: sonic hedgehog (Shh), desert hedgehog (Dhh) and indian hedgehog (Ihh). These proteins function primarily during embryonic development. Early studies have found that *Ihh* is expressed by prehypertrophic chondrocytes and that it regulates the onset of hypertrophic chondrocyte differentiation by establishing a negative feedback loop with PTHrP [30]. In a PTHrP-independent system, Ihh signaling is also required for the proper regulation of the osteoblast lineage and functions in conjunction with bone morphogenetic proteins (BMPs) to induce osteoblast differentiation from progenitor cells [6]. Additionally, Hedgehog-mediated signaling leads to activation of downstream Gli transcription factors and regulates proper osteoblast differentiation [31].

The role of Ihh in regulating chondrocyte hypertrophy has also been tested in a PTHrP-independent context using a PTHrP knockout model. In PTHrP-deficient limb explants, hypertrophic chondrocyte development is accelerated. When treated with Shh, a close analogue to Ihh, PTHrP knockout limbs display increased hypertrophic chondrocyte formation, thus demonstrating the PTHrP-independent action of Ihh to stimulate hypertrophic chondrocyte differentiation [32]. Ihh also stimulates the differentiation of periarticular to columnar chondrocytes in a PTHrP-independent manner during late embryonic development, by inducing PTHrP upregulation, elongation of columnar chondrocytes and acceleration of periarticular chondrocyte differentiation [33]. Initial studies failed to reveal the function of Ihh on postnatal bone development due to perinatal lethality of *Ihh*-deficient mutants [34]. However, inducible *cre*-based deletion of *Ihh* discovered that Ihh is essential for the maintenance of the postnatal growth plate. Loss of *Ihh* in *Col2a1*-expressing chondrocytes at a postnatal stage causes loss of the chondrocyte columnar structure, formation of ectopic hypertrophic chondrocytes and premature vascular invasion, leading to premature fusion of their growth plate and the dwarfism associated with loss of the trabecular bone over time [35]. As a whole, Ihh activation is critical for the proper development and maintenance of the postnatal growth plate and is absolutely required for chondrocyte and osteoblast differentiation.

### 4.2. The PTHrP-Ihh Negative Feedback Pathway

PTHrP was initially cloned in 1987 when discovered to be upregulated in the serum of patients with humoral hypercalcemia of malignancy, a skeletal disease associated with the incidence of bone metastases [36]. Future studies utilizing mouse and rat models discovered that *PTHrP* is expressed in the developing cartilage elements, particularly in the periarticular regions and bone collar in early bone development [37]. Therefore, *PTHrP* and *Ihh* expression patterns are coupled and work in concert with one another to either maintain embryonic and postnatal growth plate structure or promote its growth and extension.

*PTHrP* global knockout mice present an accelerated transition of proliferative chondrocytes to hypertrophic chondrocytes, resulting in premature ossification. These mice are perinatally lethal, present domed skulls, have a short snout and mandible, disproportionately short limbs and accelerated mineralization in the synchondroses of the cranial base, vertebrate and long bones [38,39,40]. The PTHrP-Ihh feedback loop is a well-accepted centerpiece that maintains growth plate structure and function. When embryonic stem cells missing both copies of the PTH/PTHrP receptor (PPR) are injected into mouse blastocysts, *PPR*-deficient mutant cells displayed premature differentiation of chondrocytes; becoming hypertrophic while surrounded by numerous normal proliferating chondrocytes, thus demonstrating the cell autonomous function of PPR in growing endochondral bones [41]. Furthermore, the lengths of columnar chondrocytes in these tissues were longer than expected, due to increased synthesis of Ihh mediated by lack of PPR in locations much closer to the ends of bones than normal. Interestingly, this ectopic *Ihh* expression was present in mutant prehypertrophic cells near the top of the growth plate. Therefore, Ihh stimulates the production of PTHrP, which in turn keeps chondrocytes proliferating and delays chondrocyte hypertrophy through a negative feedback mechanism. Through this feedback loop, PTHrP continues to be synthesized at the distal ends of the growth plate in the resting zone of the epiphyseal cartilage. When endogenous PTHrP levels are distant from the hypertrophic zone, Ihh is produced, acts on its receptor on chondrocytes to increase the rate of proliferation while stimulating the production of PTHrP [5]. These results suggest that minor changes in *PTHrP* and *Ihh* expression can be sensed by growth plate chondrocytes in a way that is dictated by the negative feedback mechanisms that determine the pace of chondrocyte differentiation in the growth plate.

Lastly, Ihh acts on perichondrial cells proximal to the hypertrophic zone to convert these cells into osteoblasts of the bone collar adjacent to the perichondrium [5]. These interactions demonstrate the importance of the PTHrP–Ihh negative feedback loop to keep chondrocytes actively proliferating while maintaining a proper balance of hypertrophic chondrocyte formation and osteoblast differentiation. Although the mechanistic details of PTHrP-Ihh interaction are not completely understood, studies have shed light on Gli2, which is the major transcription factor downstream of the Hedgehog signaling pathway [42]. *Forkhead Box C1* (*Foxc1*), a transcription factor that plays a role in early embryonic development, has a dual function in promoting proper endochondral ossification through a physical and functional interaction with Ihh-Gli2 signaling [43]. Additional investigation into the contributing factors that promote regulation of the PTHrP-Ihh negative feedback loop throughout endochondral ossification, skeletal growth and regeneration are needed.

## 5. Further Investigation of Genetic Networks That Regulate Growth Plate Chondrocytes

### 5.1. Molecular Regulation of Growth Plate Chondrocytes by a Series of Master Gene Networks

The formation of the cartilage template and subsequent osteogenic replacement of that template requires accurate and precise spatiotemporal activation of cell-signaling pathways that orchestrate the proper function of diverse cell types within the growth plate. This review will further focus on the roles of the fibroblast growth factor (FGF) family, insulin-like growth factor (IGF), Gsα-mediated signaling, Wnt signaling, runt-related transcription factor 2 (Runx2) and BMPs that potentially interact with the PTHrP-Hedgehog negative feedback mechanism to maintain growth plate activities.

### 5.2. Fibroblast Growth Factor Family and Its Relationship to Hh Activation and Chondrocyte Function

Members of the FGF family of receptors (FGFRs) and ligands have been shown in many studies to be required for proper chondrocyte function, endochondral ossification and overall skeletal development. For example, FGFRs are expressed in the postnatal growth plate, including in the perichondrium, prehypertrophic and hypertrophic zones in addition to cell types required for proper skeletal development, such as mesenchymal cells in the early craniofacial sutures [44]. Proliferative chondrocytes express *FGFR3* and pre-hypertrophic/hypertrophic chondrocytes express *FGFR1* [45,46]. Knockout of *Fgfr3* leads to an increased rate of proliferation of chondrocytes and an expansion of the length of chondrocyte columns [47]. Moreover, mutations in FGFRs cause several skeletal phenotypes such as achondroplasia, limb defects, craniosynostosis and abnormalities in phosphate regulation [44]. Importantly, FGF/FGFR signaling is a key regulator of proper bone homeostasis and therefore may play a role in bone regeneration.

Interactions between members of the FGF signaling pathways and Hedgehog family members have also been reported and shed light upon the mechanistic relationship between the two pathways. Early results showed that in response to *Fgfr3* inactivation, Hedgehog signaling is decreased, indicated by *Patched-1 (Ptch1)* expression, in proliferating chondrocytes, prehypertrophic chondrocytes and perichondrial cells [45]. However, more recent studies show that chondrocyte-specific deletion of *Fgfr3* induced multiple chondroma-like lesions, including enchondromas and osteochondromas, adjacent to disordered growth plates. These lesions also showed increased *Ihh* expression. Interestingly, when treated with an inhibitor of Hedgehog signaling, the occurrence of these lesions is significantly reduced [48]. Given the general role of FGF signaling in bone development and its potential role in Hedgehog activation, it is important to further elucidate the functional role of the FGFs during endochondral bone development.

### 5.3. Growth Factor-Stimulated Insulin-Like Growth Factor Signaling

Growth Hormone (GH) is a polypeptide hormone secreted by the anterior pituitary gland which functions primarily in the liver, where it locally stimulates IGF1 production. These molecules are endocrine factors and target a number of cell types, including cartilage and skeletal muscle [49]. Studies in *GH*-deficient animals and humans treated with IGF-1 have shown that both GH and IGF1 have the capacity to stimulate longitudinal bone growth. Moreover, *Igf1*-deficient mice show severe retardation of statural growth, first appearing during postnatal bone growth and persisting into adulthood. These animals develop dwarfism in addition to delayed ossification in cranial and facial bones [50]. Moreover, deletion of IGF1 receptor (IGF1R) in chondrocytes using *Col2a1-cre* results in perinatal lethality associated with disorganized columnar chondrocytes, delayed vascular invasion and ossification, decreased cell proliferation, increased apoptosis and increased PTHrP expression in the growth plate [51]. IGF1R may also play a role in sustaining the PTHrP-Ihh negative feedback loop during postnatal endochondral bone development, as postnatal deletion of *Igf1r* in chondrocytes results in growth retardation, disorganization of the growth plate, reduced chondrocyte differentiation, increased *PTHrP* expression and conversely, decreased *Ihh* expression [51]. In contrast, it has been also demonstrated that Ihh and IGF signaling are likely to occur independently of one another during endochondral bone development, as chondrocyte-specific deletion of Smoothened (*Smo*), a positive regulator of Hedgehog signaling, but not *Igf1r*, using *Col2a1-cre* results in a complete loss of columnar chondrocytes. [52]. Furthermore, it has been shown that the IGF1 signaling pathway may play an important role during chondrocyte hypertrophy. Hind limb specific, *Hoxb6-cre; Igf1*-deficient mice undergo normal transition into the first two phases of cell swelling; however, they do not progress through the third phase, suggesting that IGF1 is required for promoting complete hypertrophic chondrocyte formation [9]. Functional studies of the GH-IGF1 axis described here highlight the importance of the pathway in postnatal longitudinal bone growth, potentially through its interaction with the PTHrP-Ihh negative feedback loop [53].

### 5.4. Gsα-Mediated Signaling Mediates PPR Activation and Inhibits Hedgehog during Bone Development

Gsα is a physiological activator of cyclic AMP (cAMP) and protein kinase A (PKA) signaling, which has been shown to be an inhibitor of Hedgehog signaling [54]. Chondrocyte-specific deletion of *Gnas* using *Col2a1-cre* leads to perinatal lethality with severe epiphyseal growth plate abnormalities. Mutants display significant shortening of the proliferating zone and accelerated differentiation of hypertrophic chondrocytes in addition to ectopic cartilage formation at the metaphyseal edge in the tibia, due to the fact that PTHrP requires functional *Gnas* to inhibit Ihh [55]. While *PTH/PTHrP receptor* and *Ihh* expression levels in hypertrophic chondrocytes are unchanged, *Ihh*-expressing prehypertrophic chondrocytes are closer to PTHrP-secreting periarticular cells in mutant tissues. Therefore, Gnas is a direct mediator of the PTH/PTHrP receptor and PTHrP-Ihh negative feedback loop in the growth plate, which negatively regulates chondrocyte differentiation.

An additional report assessed the effects of conditionally knocking out *Gnas* in mesenchymal cells marked by *Prrx1-cre*. Mice display a phenotype similar to heterotopic ossification, characterized by ectopic bone formation in mesenchymal tissues. Furthermore, *Prrx1-cre; Gnas* deficient mice have ectopic bone formation and increased expression of osteoblast differentiation markers: Col1a1, alkaline phosphatase (Alpl) and osteocalcin (Ocn). Importantly, these mice also display significantly increased levels of Hedgehog responsive genes, *Ptch1* and *Gli1*. These results indicate that functional Gsα is required for maintaining normal levels of Hedgehog signaling via cAMP-PKA activation [56]. When Gsα is deficient, Hedgehog signaling is overactivated and promotes heterotopic ossification due to increased osteoblast differentiation.

Additional studies have identified Gsα to be the primary mediator of the actions of PTH/PTHrP receptor (PPR) in growth plate chondrocytes. *Gnas* chimeric mice lacking exon 2 (*Gnas^E2-/E2-^*) phenocopy *PPR*-deficient animals, which is indicated by premature chondrocyte hypertrophy. Furthermore, induction of a transgene expression Gsα into *Gnas^E2-/E2-^* mice prevented premature chondrocyte hypertrophy, suggesting that this exon harbors an important functional domain controlling chondrocyte differentiation [57]. Given the functional relationships between Gsα, PTHrP and Hedgehog signaling, further investigation into how these pathways regulate one another during postnatal skeletal growth and development need to be investigated.

### 5.5. Wnt Signaling Maintains the Architecture of Growth Plate Chondrocytes via Interactions with PTHrP and Ihh

Canonical Wnt signaling inhibits differentiation of early proliferating chondrocytes within the non-cartilaginous interzone during synovial joint development [58]. Ectopic activation of Wnt signaling further inhibits chondrocyte differentiation and enhances osteoblast formation in a human point mutation of lipoprotein receptor-related protein 5 (LRP5) [59]. Furthermore, genetic inactivation of β-Catenin, an essential component of canonical Wnt signaling, in mesenchymal precursor cells causes ectopic formation of chondrocytes at the expense of osteoblast differentiation both in intramembranous and endochondral ossification [60]. An additional study also shows that conditional deletion of β-Catenin in limb and head mesenchyme blocks differentiation of osteoblast precursors and diverts them into chondrocytes instead [61]. These results suggest that there is a cell-autonomous function of β-Catenin for chondrocytes and osteoblasts.

Additionally, there is substantial evidence of crosstalk between PTHrP and Wnt signaling pathways. Wnt signaling has been shown to regulate the initiation of chondrocyte hypertrophy by antagonizing PTHrP signaling, while it does not directly affect PTHrP expression. Chondrocyte-specific deletion of *β-catenin* using *Col2a1-cre* leads to a delayed onset of chondrocyte hypertrophy, which is not affected by PTHrP-deficiency. Therefore, Wnt/β-catenin signaling promotes differentiation of proliferating chondrocytes into hypertrophic chondrocytes independently of PTHrP signaling [62].

Furthermore, canonical Wnt signaling has also been shown to interact with Ihh in PTHrP-independent situations. Ihh activation may be upstream of Wnt signaling in the perichondrium, as *Ihh*-deficient mice demonstrate loss of *Wnt7b* and *Tcf1*, which are two indicators of canonical Wnt signaling, in the perichondrium [63]. Sox9 physically and functionally interacts with β-catenin [64]. Furthermore, chondrocyte-specific deletion of *β-catenin* and overexpression of Sox9 display similar phenotypes, resulting in hypoplastic and shortened endochondral bones, whereas chondrocyte-specific deletion of *Sox9* and constitutive activation of *β-catenin* also display similar phenotypes, resulting in generalized chondrodysplasia. These functional in vivo experiments underscore the negative relationship of Sox9 and β-catenin during endochondral bone formation and chondrogenesis that cooperatively regulate chondrocyte proliferation and differentiation. Additionally, loss of *Wnt9a* could temporally and spatially downregulate *Ihh* signaling in the appendicular skeleton leading to a delay in chondrocyte and osteoblast maturation as well as shortening of the proximal long bones [65]. In this study, β-catenin and lymphoid enhancer binding factor 1 (Lef1) have been shown to associate with the *Ihh* promoter in vivo.

Importantly, non-canonical Wnt signaling has also been shown to play a role in maintenance of the growth plate. For example, Wnt5a and Wnt5b regulate the transition between chondrocyte zones in the postnatal growth plate, independently of the PTHrP-Ihh negative feedback loop [66]. Given these extensive data, non-canonical and canonical Wnt signaling plays an important role in promoting proper skeletal growth and development via interactions with PTHrP and Ihh-dependent and independent biological contexts [67].

### 5.6. Runx2 Activation Is Required for Initial Bone Formation and Osteoblast Function

Runx2 is a master regulator of osteoblast differentiation. *Runx2*-deficient mice lack osteoblasts and exhibit abnormalities in hypertrophic chondrocytes. Moreover, these limited hypertrophic chondrocytes are unable to undergo mineralization associated with decreased levels of osteopontin (Opn) and matrix metalloproteinase 13 (Mmp13) [68,69,70]. Conversely, transgenic overexpression of *Runx2* in *Col2a1*-expressing cells results in accelerated chondrocyte hypertrophy and induces mineralization in cartilage that does not undergo osteogenesis normally [70]. Furthermore, *Runx2* expression is first detected in hypertrophic chondrocytes, including terminally differentiated hypertrophic chondrocytes [71]. Importantly, Runx2 activates *Ihh* expression [72]. Runx2 plays a fundamental role in maintaining the distribution of the zones of the growth plate and is a known critical regulator of osteoblast and chondrocyte differentiation.

### 5.7. Bone Morphogenetic Protein Pathway and Its Functional Role in Bone Development

The BMP signaling pathway includes members of the transforming growth factor-beta (TGFβ) superfamily of proteins, a family that has been shown to play an important role in proper skeletal development and bone formation [73,74]. More specifically, BMP signal activation is required for undifferentiated mesenchymal cells to become precursors of osteoblasts and chondrocytes. For example, an early study shows that mice deficient for *Gdf5*, a closely related member of the TGFβ/BMP family, display brachypodism [75]. Furthermore, double conditional knockout of Bmp2 and Bmp4 in *Col2a1*-expressing chondrocytes results in disorganization of chondrocytes, decreased cell proliferation, poor differentiation and increased apoptosis in the growth plate [76]. An additional study assessed the skeletal viability of a *Prrx1-cre; Bmp2* model. Although mutant animals have few skeletal abnormalities at birth, they prematurely develop the secondary ossification center shortly thereafter in several bone epiphyses. *Prrx1-cre; Bmp2* mice also have an increased incidence of spontaneous limb fractures, resulting in significantly impaired locomotion. In response to long bone fracture, *Bmp2*-deficient animals fail to undergo fracture healing due to the absence of chondrogenesis at the sites of injury [77]. Multiple other members of the BMP pathway, including downstream Smad proteins, have defects in appendicular skeletal development when conditionally ablated in chondrocytes [78]. Proper activation of BMP signaling is required for the differentiation of pre-chondrocytes, underlying its functional importance in regulating initial bone development and postnatal skeletal growth [77].

The BMP signaling pathway has also been shown to directly interact with Ihh and PTHrP. Ihh signaling is directly required for BMP-induced osteoblastogenesis in vitro [6]. Additional studies have found that BMP signaling interacts with Hedgehog signaling to enhance bone collar formation. However, in the absence of Hedgehog input, BMP signaling increased ectopic chondrogenesis in the perichondrium [79]. BMP2 also induces expression of *Ptch1*, during osteoblast differentiation in a positive feedback loop mechanism [6]. Studies have also shown that BMP signaling functions within the PTHrP-Ihh negative feedback loop by bolstering this regulatory mechanism, working in conjunction with Ihh to maintain a normal chondrocyte proliferation rate [80,81]. Given these reports, BMP signaling is required for normal osteogenesis and plays a significant role in promoting proper endochondral bone formation via its interactions with the PTHrP-Ihh negative feedback loop [82].

## 6. Conclusions

Chondrocytes represent a unique cell type that can undergo dynamic morphological changes along their differentiation trajectory during normal development. Recent studies demonstrate that a stem cell niche exists in the postnatal growth plate, in which special chondrocytes undergo self-renewal and support extensive bone growth and elongation. Chondrocytes are regulated by complex and multifactorial molecular signaling pathways, which govern their ability to self-renew and differentiate into diverse cell types, such as columnar chondrocytes in the growth plate as well as osteoblasts and bone marrow stromal cells in the primary spongiosa and marrow cavity (Figure 1). Given the scientific era where we observe explosive technological advances on a daily basis, the application of single cell RNA transcriptomics and other omic-based technologies will enable researchers to further elucidate the fundamental molecular properties of individual chondrocytes in a spatiotemporal manner during various stages of skeletal development [83,84]. By doing so, we become one step closer to bridging the gap between the clinic and the bench, using novel regenerative therapies for patients suffering from congenital skeletal defects or those who have suffered through traumatic injuries to the skeleton.

## Figures and Tables

**Figure 1 ijms-20-06009-f001:**
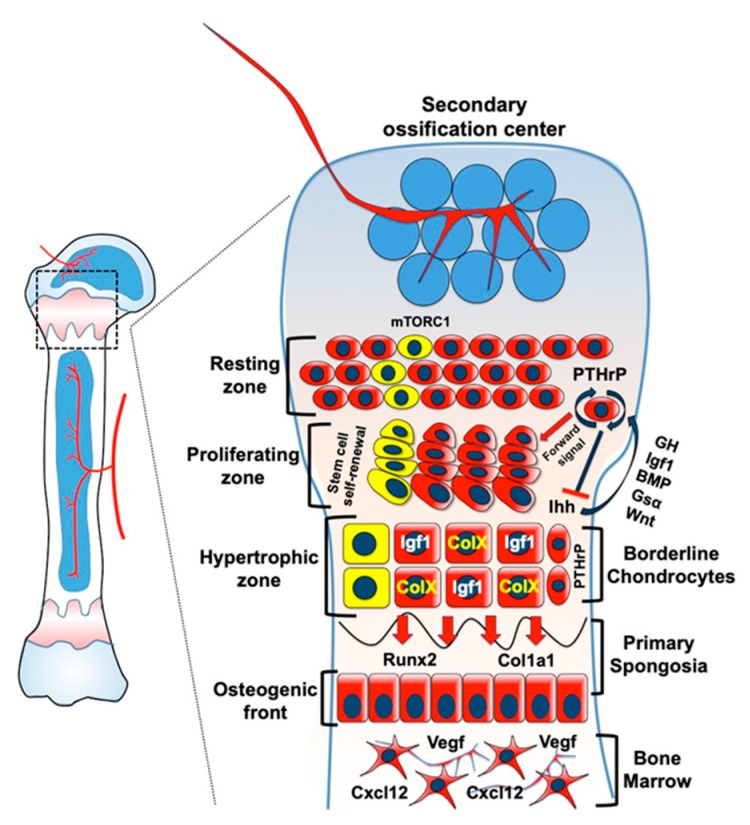
Molecular characteristics of growth plate chondrocytes and their differentiation trajectories. The initial development of the secondary ossification center stimulates the formation of a PTHrP-labeled skeletal stem in the resting zone niche. This population contributes to the formation of proliferating and hypertrophic chondrocytes in addition to marking Col1a1^+^ osteoblasts in the primary spongiosa and Cxcl12^+^ bone marrow stromal cells in the marrow cavity over time. Moreover, the secondary ossification center promotes the establishment of a stem cell niche in the postnatal epiphyseal plate that is maintained by mTORC1 activity. Importantly, a paradigm has established that hypertrophic chondrocytes have the ability to ‘transdifferentiate’ into osteoblasts and further into more mature osteocytes over time, suggesting that this cell type, while terminally differentiated, does not always undergo apoptosis.

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
