# Peer review of "Growth Plate Chondrocytes: Skeletal Development, Growth and Beyond"

_ijms, 2019, doi:10.3390/ijms20236009_

Round 1

Reviewer 1 Report

This is a well written review. It reviews the current knowledge about the growth plate very well.

Author Response

Response to Reviewer #1

Thank you very much for your valuable comments and critiques

<Reviewer>

This is a well written review. It reviews the current knowledge about the growth plate very well.

<Response>

Thank you for your kind response!

Reviewer 2 Report

This is a review article covering broad topics related to growth plate biology.  It is challenging to discuss each topic citing key papers from the vast literature on growth plates.   

Specific comments:

 ln 53. articular cartilage develops from a different cell origin and through a diffenent mechanism from those of growth plate chondrocytes.  See the paper, Kozhemyakina E et al. 2015. PMID:25603997  ln 196.  Osx is not genetically upstream of Runx2.  Also official gene nomenclature should be used.   ln 213  literature 24 describes growth palte phenotype of Col10 mutant mice, which can not be generalized like the statement of this sentence. ln 288  Pls cite literature that provides evidence for the statment "Ihh activation is critical for the proper maintenance of the POSTNATAL gtrowth plate,.." Section 5.3. ln 373.  The paper with the opposite conclusion to this statment should be also cited: Long F et al 2006 PMID:16905129. Section 5.4. the paper by Sakamoto et al 2005 PMID: 15765186 should be cited. Section 5.5 ln 406 states that Canonical Wnt signaling inhibit chondrocyte differentiation..."  This statement is ambiguous as it does not specify what stage of differentiation of chondrocytes.  As said in ln 416, canonical Wnt promotes chondrocyte differentiation to hypertrophy.  Pls make the statement more accurate and unambigous.  The paper by Akiyama et al 2004PMID:15132997 should be cited. ln 418 Wnt 5a 5b.  Non canonical Wnts should be separately discussed from canonical Wnts.   ln423  Wnt9a but not Wnt9. ln 446  Gdf5 but not Bmp5. Minor rammatical errors should be corrected. 

Author Response

Response to Reviewer #2

Thank you very much for your valuable comments and critiques. We have addressed your comments and made the revision as described in the following.

<Reviewer>

ln 53. articular cartilage develops from a different cell origin and through a different mechanism from those of growth plate chondrocytes. See the paper, Kozhemyakina E et al. 2015. PMID:25603997

<Response>

Following the reviewer’s suggestion, we have now added the following sentence in the main text (Line 56-58).

The following sentence was inserted into the main text (Section 2.1):

The articular cartilage develops from a different cell origin and through a different mechanism from those of growth plate chondrocytes [12].

<Reviewer>

ln 196. Osx is not genetically upstream of Runx2. Also official gene nomenclature should be used.

<Response>

We have revised this error in the revised manuscript. We have also added the official gene nomenclature for Osx (i.e. Sp7) in the revised manuscript.

The following underlined word was revised in the main text (Section 3.3):

For example, Sp7 (Osx) is genetically downstream of Runx2, …

<Reviewer>

            ln 213 literature 24 describes growth plate phenotype of Col10 mutant mice, which can not be generalized like the statement of this sentence.

<Response>

            Following the reviewer’s suggestion, we have clarified this point in the revised manuscript.

The following underlined clauses were inserted into the main text (Section 3.3):

            Transgenic overexpression of mutant Col10a1 results in endoplasmic reticulum stress and the failure of hypertrophic chondrocytes to terminally differentiate, resulting in delayed endochondral bone formation and a chondrodysplasia phenotype.

<Reviewer>

            ln 288 Pls cite literature that provides evidence for the statement "Ihh activation is critical for the proper maintenance of the POSTNATAL growth plate,.."

<Response>

            We have revised the main text to meet the reviewer’s request. More specifically, we have cited a more recent report by Maeda et al. regarding Ihh function on postnatal long development.

The following sentences were added to the main text (Section 4.1):

            Initial studies failed to reveal the function of Ihh on postnatal bone development due to perinatal lethality of Ihh-deficient mutants [34]. However, inducible cre-based deletion of Ihh discovered that Ihh is essential for the maintenance of the postnatal growth plate. Loss of Ihh in Col2a1-expressing chondrocytes at a postnatal stage causes loss of the chondrocyte columnar structure, formation of ectopic hypertrophic chondrocytes and premature vascular invasion, leading to premature fusion of their growth plate and the dwarfism associated with loss of the trabecular bone over time [35].

St-Jacques, B.; Hammerschmidt, M.; McMahon, A.P. Indian hedgehog signaling regulates proliferation and differentiation of chondrocytes and is essential for bone formation. Genes Dev 1999, 13, 2072-2086, doi:10.1101/gad.13.16.2072. Maeda, Y.; Nakamura, E.; Nguyen, M.T.; Suva, L.J.; Swain, F.L.; Razzaque, M.S.; Mackem, S.; Lanske, B. Indian Hedgehog produced by postnatal chondrocytes is essential for maintaining a growth plate and trabecular bone. Proc Natl Acad Sci U S A 2007, 104, 6382-6387, doi:10.1073/pnas.0608449104.

<Reviewer>

            Section 5.3. ln 373. The paper with the opposite conclusion to this statement should be also cited: Long F et al 2006 PMID:16905129.

<Response>

            Following the reviewer’s suggestion, we have added the following sentence in the main text to mention the Long paper in 2006.

The following sentence was added to the main text (Section 5.3):

            In contrast, it has been also demonstrated that Ihh and IGF signaling are likely to occur independently of one another during endochondral bone development, as chondrocyte-specific deletion of Smo, but not Igf1r, using Col2a1-cre results in a complete loss of columnar chondrocytes. [52].

Long, F.; Joeng, K.S.; Xuan, S.; Efstratiadis, A.; McMahon, A.P. Independent regulation of skeletal growth by Ihh and IGF signaling. Dev Biol 2006, 298, 327-333, doi:10.1016/j.ydbio.2006.06.042.

<Reviewer>

Section 5.4. the paper by Sakamoto et al 2005 PMID: 15765186 should be cited.

<Response>

            Following the reviewer’s suggestion, we have added the following sentence in the main text to mention the Sakamoto paper in 2005.

The following sentences were added to the main text (Section 5.4):

            Chondrocyte-specific deletion of Gnas using Col2a1-cre leads to perinatal lethality with severe epiphyseal growth plate abnormalities. Mutants display significant shortening of the proliferating zone and accelerated differentiation of hypertrophic chondrocytes in addition to ectopic cartilage formation at the metaphyseal edge in the tibia, due to the fact that PTHrP requires functional Gnas to inhibit Ihh [58]. While PTH/PTHrP receptor and Ihh expression levels in hypertrophic chondrocytes are unchanged, Ihh-expressing prehypertrophic chondrocytes are closer to PTHrP-secreting periarticular cells in mutant tissues. Therefore, Gnas is a direct mediator of the PTH/PTHrP receptor and PTHrP-Ihh negative feedback loop in the growth plate, which negatively regulates chondrocyte differentiation.

Sakamoto, A.; Chen, M.; Kobayashi, T.; Kronenberg, H.M.; Weinstein, L.S. Chondrocyte-specific knockout of the G protein G(s)alpha leads to epiphyseal and growth plate abnormalities and ectopic chondrocyte formation. J Bone Miner Res 2005, 20, 663-671, doi:10.1359/JBMR.041210.

<Reviewer>

            Section 5.5 ln 406 states that Canonical Wnt signaling inhibit chondrocyte differentiation..." This statement is ambiguous as it does not specify what stage of differentiation of chondrocytes.

<Response>

            Following the reviewer’s suggestion, we have now revised this sentence for clarification.

The following underlined clauses were added to the main text (Section 5.5):

            Canonical Wnt signaling inhibits differentiation of early proliferating chondrocytes within the non-cartilaginous interzone during postnatal synovial joint development [61].

<Reviewer>

            As said in ln 416, canonical Wnt promotes chondrocyte differentiation to hypertrophy. Pls make the statement more accurate and unambigous.

<Response>

            Following the reviewer’s suggestion, we have now revised the sentence for clarification.

The following sentences were added to the main text (Section 5.5):

            Chondrocyte-specific deletion of β-catenin using Col2a1-cre leads to a delayed onset of chondrocyte hypertrophy, which is not affected by PTHrP-deficiency. Therefore, Wnt/β-catenin signaling promotes differentiation of proliferating chondrocytes into hypertrophic chondrocytes independently of PTHrP signaling [65].

<Reviewer>

            The paper by Akiyama et al 2004PMID:15132997 should be cited.

<Response>

            Following the reviewer’s suggestion, we have added the following sentence in the main text to mention the Akiyama paper in 2004.

The following sentences were added to the main text (Section 5.5):

            Sox9 physically and functionally interacts with β-catenin [68]. Furthermore, chondrocyte-specific deletion of β-catenin and overexpression of Sox9 display a similar phenotype, resulting in hypoplastic and shortened endochondral bones, whereas chondrocyte-specific deletion of Sox9 and constitutive activation of β-catenin also display similar phenotypes, resulting in generalized chondrodysplasia. These functional in vivo experiments underscore the negative relationship of Sox9 and β-catenin during endochondral bone formation and chondrogenesis that cooperative regulate chondrocyte proliferation and differentiation.

Akiyama, H.; Lyons, J.P.; Mori-Akiyama, Y.; Yang, X.; Zhang, R.; Zhang, Z.; Deng, J.M.; Taketo, M.M.; Nakamura, T.; Behringer, R.R., et al. Interactions between Sox9 and beta-catenin control chondrocyte differentiation. Genes Dev 2004, 18, 1072-1087, doi:10.1101/gad.1171104.

<Reviewer>

            ln 418 Wnt 5a 5b. Non canonical Wnts should be separately discussed from canonical Wnts.

<Response>

            Following the reviewer’s suggestion, we have now discussed non-canonical Wnt studies separately, at the end of the section.

The following underlined sentence was added to the main text; the paragraph was moved at the end of (Section 5.5):

            Importantly, non-canonical Wnt signaling has also been shown to play a role in maintenance of the growth plate. For example, Wnt5a and Wnt5b regulate the transition between chondrocyte zones in the postnatal growth plate, independently of the PTHrP-Ihh negative feedback loop [66].

<Reviewer>

            ln423 Wnt9a but not Wnt9.

<Response>

            We have revised this error in the revised manuscript, thank you very much.

<Reviewer>

            ln 446 Gdf5 but not Bmp5.

<Response>

            We have revised this error in the revised manuscript with additional clarification, thank you very much.

The following underlined clause was added to the main text (Section 5.7):

…an early study shows that mice deficient for Gdf5, a closely related member of the TGFβ/BMP family, display brachypodism [78].

<Reviewer>

            Minor grammatical errors should be corrected.

<Response>

            We have scrutinized the main text and addressed minor grammatical errors in the revised text.